# 3D ECM-Based Scaffolds Boost Young Cell Secretome-Derived EV Rejuvenating Effects in Senescent Cells

**DOI:** 10.3390/ijms24098285

**Published:** 2023-05-05

**Authors:** Sharon Arcuri, Georgia Pennarossa, Teresina De Iorio, Fulvio Gandolfi, Tiziana A. L. Brevini

**Affiliations:** 1Laboratory of Biomedical Embryology and Tissue Engineering, Department of Veterinary Medicine and Animal Sciences, Center for Stem Cell Research, Università degli Studi di Milano, Via Trentacoste 2, 20134 Milan, Italy; sharon.arcuri@unimi.it (S.A.); teresina.deiorio@unimi.it (T.D.I.); 2Department of Agricultural and Environmental Sciences-Production, Landscape, Agroenergy, Università degli Studi di Milano, Via Celoria 2, 20133 Milan, Italy; fulvio.gandolfi@unimi.it

**Keywords:** aging, cellular rejuvenation, ECM-based bio-scaffolds, EVs, cell senescence, young, secretome

## Abstract

Aging is a complex, multifaceted degenerative process characterized by a progressive accumulation of macroscopic and microscopic modifications that cause a gradual decline of physiological functions. During the last few years, strategies to ease and counteract senescence or even rejuvenate cells and tissues were proposed. Here we investigate whether young cell secretome-derived extracellular vesicles (EVs) ameliorate the cellular and physiological hallmarks of aging in senescent cells. In addition, based on the assumption that extracellular matrix (ECM) provides biomechanical stimuli, directly influencing cell behavior, we examine whether ECM-based bio-scaffolds, obtained from decellularized ovaries of young swine, stably maintain the rejuvenated phenotype acquired by cells after exposure to young cell secretome. The results obtained demonstrate that young cells release EVs endowed with the ability to counteract aging. In addition, comparison between young and aged cell secretomes shows a significantly higher miR-200 content in EVs produced using fibroblasts isolated from young donors. The effect exerted by young cell secretome-derived EVs is transient, but can be stabilized using a young ECM microenvironment. This finding indicates a synergistic interaction occurring among molecular effectors and ECM-derived stimuli that cooperate to control a unique program, driving the cell clock. The model described in this paper may represent a useful tool to finely dissect the complex regulations and multiple biochemical and biomechanical cues driving cellular biological age.

## 1. Introduction

Human life expectancy is increasing worldwide at a rapid rate thanks to improvements in medical care, healthier lifestyles, and a significant reduction in child mortality [1]. Although this reflects a positive development, it also poses new challenges that will aggravate in the coming years. Indeed, the increase in lifespans of the last decades was not paralleled with enhancements in life quality for the elderly, and the risk of chronic age-onset diseases, often with multiple co-morbidities, is steadily on the rise, thus becoming one of the most critical emerging issues in public health [2].

To overcome these problems, several recent studies focused on in-depth characterization of aging, leading to the identification of different molecular and biomechanical mechanisms that drive and/or influence senescence progression [3,4,5,6,7,8,9,10]. Based on this new knowledge, strategies to ease, stop, or counteract the accumulation of macroscopic and microscopic modifications that are distinctive of age progression and negatively affect organ, tissue, cell, and subcellular organelle homeostasis and functions are proposed [8]. Some of these methods are based on nutrient-related pathway alterations, either through dietary restrictions [11]; pharmacological interventions employing chemical drugs, such as rapamycin, metformin, or resveratrol [12]; or the use of individual “rejuvenating” factors, such as Growth Differentiation Factor 11 (GDF11), Tissue Inhibitor of Metalloproteinases 2 (TIMP2), and Mesencephalic Astrocyte Derived Neurotrophic Factor (MANF) [12,13]. Other approaches involve the induction of a stable [14] or transient [3] pluripotent state for ameliorating cellular and physiological aging hallmarks in senescent or centenarian cells. Parallel reports also describe the possibility of reverting aging-related features in old stem cells through culturing them with young stem cell-derived conditioned medium, also referred as “secretome” [11,15,16,17]. This method is composed of a complex set of soluble bio-active molecules released by cells, including serum proteins, angiogenic and growth factors, hormones, cytokines, extracellular matrix proteins, and extracellular vesicles (EVs) [18]. EVs are lipid bound structures that contain proteins, lipids, nucleic acids, and metabolites [19,20]. They are known to facilitate intercellular communications and play a key role in a variety of physiological and pathological processes, including immuno-regulation, cell differentiation and metabolism, and cancer and autoimmune diseases [19,20].

In the present study, we investigate whether young cell secretome-derived EVs ameliorate cellular and physiological hallmarks of aging in senescent cells. Taking advantage of acellular scaffold low immunogenicity, we use decellularized bio-scaffolds obtained from young pigs to examine whether extracellular matrix (ECM)-based supports can boost and properly maintain the rejuvenated phenotype acquired by senescent cells after exposure to young cell secretome-derived EVs. In addition, based on previous observations indicating the miR-200 family’s ability to regulate the molecular mechanisms driving cellular senescence erasure [3,21], we compare young and aged cell secretome-derived EVs for their content in miR-200b and miR-200c.

## 2. Results

### 2.1. Young Cell Secretome-Derived EVs Transiently Erase Signs of Senescence in Fibroblasts Isolated from Aged Individuals Cultured in 2D Systems

After 48-h exposure to young cell conditioned media (Post treatment yCM), fibroblasts isolated from aged patients showed phenotype changes. The elongated morphology, visible in untreated cells (Aged, Figure 1A), was replaced with a rounder and more oval shape, with cells becoming smaller in size (yCM, Figure 1A). Comparable phenotype modifications were observed when aged cells were treated with yCM-derived EVs resuspended in Eagle’s Minimum Essential Medium (MEM) (Post treatment MEM + yCM-derived EVs Figure 1A). In contrast, fibroblasts maintained the original phenotype when incubated with EV-depleted yCM (Post EV-yCM, Figure 1A) or aged cell conditioned media (Post aCM, Figure 1A).

The morphological changes visible after exposure to yCM and MEM + yCM-derived EVs were accompanied with a significant decrease in β-galactosidase (β-GAL) activity (post-treatment yCM and MEM + yCM-derived EVs, Figure 1B), as well as significantly lower levels of reactive oxygen species (ROS) (post-treatment yCM and MEM + yCM-derived EVs, Figure 1C), with values comparable to those detected in young cells (Young, Figure 1B,C). In contrast, EV-yCM and aCM treatments did not induce β-GAL and ROS level changes, and their values remained statistically comparable to those of untreated aged fibroblasts (Aged, Figure 1B,C).

Gene expression analyses were consistent with the morphological observations and demonstrated a significant decrease in the transcription levels of senescence-related markers (P53, P16 and P21) and the reactive oxygen species modulator ROMO1 after 48 h of treatment with both yCM (Post treatment yCM) and MEM + yCM-derived EVs (post-treatment MEM + yCM-derived EVs, Figure 2), showing values comparable to those distinctive of cells isolated from young donors (Young). In addition, expression levels of the cell proliferation marker MKI67, as well as the mitochondrial activity-related genes TFAM, PDHA1, and COX4l1, peaked at values comparable to those of fibroblasts derived from young individuals (Young, Figure 2).

PCNA immunocytochemical analysis and its quantitative evaluation demonstrated a significantly higher PCNA positive cell rate after yCM (post-treatment yCM, Figure 3A,B) and MEM + yCM-derived EVs (post-treatment MEM + yCM-derived EVs, Figure 3A,B) exposure, while no differences were detected among untreated aged fibroblasts (Aged, Figure 3A,B) and cells incubated with EV-yCM (post-EV-yCM, Figure 3A,B) or aCM (Post aCM, Figure 3A,B).

However, when yCM and MEM + yCM-derived EV treated cells were returned to fibroblast standard culture medium, they progressively reverted to their original phenotype (Day 2, day 5 and day 7; Figure 1A); by day 10, they exhibited small central nuclei and an elongated spindle shape, comparable to characteristics of untreated aged fibroblasts (Aged, Figure 1A). Consistently, β-GAL activity (Figure 1B) and ROS levels (Figure 1C) gradually incremented during the culture period up to day 10, when the values detected were statistically comparable to those observed in untreated aged cells (Aged). These changes were paralleled by P53, P16, P21, and ROMO1 gene transcription levels that gradually increased and returned results statistically comparable to those of untreated aged cells (Aged) by day 10 of culture (Figure 2). Similarly, by day 10, MKI67, TFAM, PDHA1, and COX4I1 genes decreased to expression levels comparable to those detected in aged fibroblasts (Aged, Figure 2). Immunostaining confirmed these data, with PCNA+ cell rates slowly decreasing and returning results statistically comparable to those of aged cells by day 10 of culture (Figure 3A,B).

### 2.2. Young Cell-Derived EV Rejuvenating Effects Are Boosted and Steadly Maintened in Cells Grown onto Young 3D ECM-Based Bio-Scaffolds

H and E and DAPI staining (Figure 4A) showed comparable engrafting ability in both yCM and EV-yCM treated cells, demonstrating that EV exposure does not affect cell repopulating ability. However, although cell density analyses indicated an increasing number of cells along the culture period in all the experimental groups, significant differences were visible (Figure 4B). Moreover, from day 2 onward, cell density was significantly higher in yCM cultured onto young decellularized bio-scaffolds (yCM + Young scaffold, Figure 4B) compared to EV-yCM engrafted onto young decellularized bio-scaffolds (EV-yCM + Young scaffold, Figure 3B) and yCM cultured onto aged decellularized bio-scaffolds (yCM + Aged scaffold, Figure 3B). These histological observations were also confirmed through DNA quantification studies, indicating comparable cell growth trends (Figure 4C).

Gene expression studies indicated that the significant reductions in the transcription levels of P53, P16, P21, and ROMO1 genes detected after 48-h exposure to yCM (Post treatment yCM, Figure 5) were steadily maintained with values statistically comparable to those distinctive of young cells (Young, Figure 5), albeit only when treated cells were engrafted onto young scaffolds (yCM + Young scaffold, Figure 5). Similarly, the increased expression values of MKI67, TFAM, PDHA1, and COX4I1 genes visible after yCM incubation (Post treatment yCM, Figure 5) remained statistically, albeit only comparable to those of young cells in the yCM + Young scaffold group (Figure 5).

In agreement with this, the decreased β-GAL activity and ROS levels, which were visible after 48-h incubation with yCM (Post treatment yCM, Figure 6A,B), remained statistically comparable to those distinctive of young cells (Young), albeit only when treated cells were cultured onto young scaffolds (yCM + Young scaffold, Figure 6A,B). In addition, PCNA staining revealed immunopositivity in all experimental groups (Figure 6C). However, a significantly higher number of PCNA+ cells was scored in young scaffolds repopulated with yCM-treated cells (yCM + Young scaffold, Figure 6D) compared to the other groups (Figure 6D).

### 2.3. Young and Aged Cell Secretome-Derived EVs

TEM studies demonstrated the presence of polydisperse spherical structures ranging from 30 to 250 nm in size (Figure 7A) and displaying the distinctive EV cup morphology, surrounded by a lipid bilayer with variable electron density cargo content. Consistent with this result, WB analysis indicated that both yCM- and aCM-derived EVs contained the tetraspanin markers CD9, CD63, and CD81 at similar levels, when equal amounts of EV proteins were loaded (Figure 7B). However, EV quantification studies revealed that aCM contained a significantly higher amount of EVs than yCM (Figure 7C). Nevertheless, an opposite trend was observed when assessing miRNA concentrations in yCM- and aCM-derived EVs, with significantly higher amounts of miR-200b and miR-200c in yCM-derived EVs (Figure 7D).

## 3. Discussion

In the present study, we demonstrate that young cells release EVs endowed with the ability to counteract aging. Interestingly, this effect is transient; however, it can be stabilized using the contribution derived from young ECM microenvironment. We also show that young cell secretome-derived EVs contain a significantly higher miR-200 amount, compared to those EVs isolated from aged donors.

After yCM exposure, fibroblasts obtained from aged individuals acquire morphological and molecular features distinctive of cells derived from young donors. Moreover, at the end of 48-h treatment with yCM or MEM + yCM-derived EVs, the fibroblast elongated shape is replaced by a more compact and rounded phenotype, with cells becoming smaller in size. In contrast, when cells are exposed to EV-yCM or aCM, no changes are detected, and cell morphology remains comparable to that of untreated cells. These results clearly indicate that EVs released by young fibroblasts directly influenced cellular remodeling. Morphological observations are also supported through functional and molecular analyses, which demonstrate a significant reduction in the main age-related hallmarks in cells treated with yCM and MEM + yCM-derived EV. In particular, exposure of aged cells to young cell secretome-derived EVs induces a significant decrement in β-GAL and ROS activities, with values comparable to those detected in young cells. This result is paralleled through a statistically significant downregulation of P53, P21, P16, and ROMO1 gene transcription, and increased expression levels for the mitochondrial- (TFAM, PDHA1, and COX4I1) and proliferation-related (MKI67) genes. In contrast, EV-yCM and aCM treatments neither induce β-GAL and ROS level changes nor transcriptional activity modifications, with all values remaining statistically comparable to those of untreated aged fibroblasts. This result agrees with recent studies demonstrating young cell derived- EV ability to mediate senescence rejuvenation [11,22,23,24]. In particular, the authors report the possibility of reducing signs of aging in a variety of tissues in old mice [23], as well as improving wheel-running activity and extending lifespans in aged mice exposed to EVs isolated from young individuals [22]. In addition, Fafián-Labora et al. also showed that EVs isolated from primary fibroblasts of young donors ameliorate certain senescence-related biomarkers in cells derived from old and Hutchinson–Gilford progeria syndrome donors [23].

The age reversion observed in the present study is also confirmed via PCNA immunocytochemical analysis and its quantitative evaluation, which demonstrate a significantly higher PCNA+ cell rate after yCM and MEM + yCM-derived EV exposure, and no differences detected in aged cells incubated with EV-yCM or aCM. These data suggest the ability of EVs derived from young cells to re-activate cell-cycle and re-establish a robust cell division rate in senescent cells. Consistent with this result, recent pilot works described the possibility of achieving a young phenotype after restoration of a vigorous cell growth in non-dividing quiescent and senescent cells [4,5], pointing to a scenario where proliferation was an essential requirement for cellular rejuvenation [6]. Altogether, these data indicate the key role played by young cell secretome-derived EVs in reducing signs of cellular senescence and encouraging the acquisition of a young phenotype. It is, however, important to note that, when yCM or MEM + yCM-derived EVs are removed from cultures and cells are returned to fibroblast standard culture medium, a gradual reversion to the original phenotype is detected, with cells restoring all the morphological and molecular features distinctive of aged cells by day 10 of culture. Specifically, cells revert to a larger size and have an elongated shape, increased β-GAL and ROS activities, and MKI67, TFAM, PDHA1, COX4I1, ROMO1, P53, P16, and P21 transcription levels statistically comparable to those of untreated aged cells.

While the rejuvenating effect exerted through young cell secretome-derived EVs appears to be transient and reversible, it is important to note that, in combination with the use of young 3D ECM-based bio-scaffolds, it results in a stable acquisition of a young phenotype. Indeed, even after the removal of young cell secretome-derived EVs, cells cultured in fibroblast standard culture medium continue to colonize the bio-scaffolds, and stably maintain all the distinctive features of young cells. They display low β-GAL and ROS values, a high number of PCNA+ cells, high transcription levels for cell proliferation- (MKI67) and mitochondrial activity-related genes (TFAM, PDHA1, and COX4I1), and low expression of P53, P16, P21, and ROMO1. These results confirm that an adequate young microenvironment may stabilize and support the rejuvenated phenotype acquired by aged cells in response to anti-aging factors and suggests the involvement of synergistic interactions among soluble effectors and ECM-derived stimuli [3].

To aid our understanding, a key means of further elucidating the contribution of soluble effectors was to analyze and compare the EVs isolated from young and aged cell secretomes. TEM studies demonstrated the presence of polydisperse spherical structures with distinctive EV size and morphology in both yCM and aCM. Similarly, WB analysis indicated that the tetraspanin markers CD9, CD63, and CD81 were contained at similar levels in both the experimental groups, when equal amounts of EV proteins were loaded. This result suggests that both young and senescent cells release EVs in the culture medium. However, EV quantification studies reveal that aCM contains a significantly higher amount of EVs than yCM. Although further experiments are needed, this result is consistent with recent studies demonstrating an increase in EV secretion and a modification in their bioactive content connected with cellular aging, thus resulting in a pattern that constitutes the senescence-associated secretory phenotype (SASP) [11,25,26,27,28,29,30,31]. On the other hand, it is important to note that several reports have highlighted variations in EV-containing molecules in relation to cellular age [25,26,27]. Based on this factor and recent evidence obtained in our laboratory demonstrating the miR-200 family’s ability to transiently rejuvenate senescent cells [3], we investigated miR-200 contents in young and aged cell secretome-derived EVs. The results obtained demonstrate that yCM-derived EVs have significantly higher amounts of miR-200b and miR-200c compared to aCM-derived EVs, suggesting the intriguing possibility that miR-200 family may be one of the paracrine effectors involved in cellular rejuvenation. In agreement with this hypothesis, the miR-200 family’s ability to restore normal functions in different senescent cell types was previously demonstrated [3,21]. A clear example can be found in idiopathic pulmonary fibrosis, which causes downregulation of miR-200 members in alveolar epithelial cells, leading to the acquisition of a senescent phenotype [32,33]. It is, however, interesting to note that miR-200b and miR-200c transfection restores cell trans-differentiation ability and induces a significant reduction in aging hallmarks in senescence alveolar cells [21]. Similarly, the use of miR-200b and miR-200c was shown to directly regulate the molecular mechanisms erasing cell senescence in fibroblasts isolated from old individuals [3]. Although further studies are needed to better elucidate these aspects, based on these observations, we may speculate that young cells release EVs containing high amount of miR-200s, which are required to support normal physiological cell functions, and that this ability decreases in aged and pathological cells.

In conclusion, the data reported in this study demonstrate that multiple factors cooperate to modulate the cell clock. In particular, the paracrine effectors released by young cells appear to play a fundamental role in erasing cellular senescence. However, the combination of their action with young ECM-derived stimuli boosts and stabilize the anti-aging effect observed in the experimental model used, which may represent a useful tool to finely dissect the complex regulations driving cellular biological age.

## 4. Materials and Methods

All reagents were purchased from Thermo Fisher Scientific, Milan, Italy, unless otherwise indicated.

### 4.1. Ethical Statement

Human primary skin fibroblast cell lines (GM02674, GM00495, GM08402, GM01706, GM00731, and AG09602) were obtained from the NIGMS Human Genetic Cell Repository, based at the Coriell Institute for Medical Research (Camden, NJ, USA). Porcine ovaries were collected from an authorized local slaughterhouse. This study did not involve the use living humans and animals; therefore, ethical approval was not required. All the methods were carried out following the approved guidelines.

### 4.2. Culture of Human Skin Fibroblasts

Fibroblasts isolated from 29–32 years old individuals (young, *n*  =  3) and from 82–96 year-old donors (aged, *n*  =  3) were cultured in fibroblast standard culture medium (FCM) consisting of Eagle’s Minimum Essential Medium (MEM), 15% Fetal Bovine Serum (FBS, not heat-inactivated), 2mM glutamine (Sigma-Aldrich, Milan, Italy), and 1% antibiotic/antimycotic solution (Sigma-Aldrich, Milan, Italy). Cells were maintained in 5% CO_2_ at 37 °C and passaged twice per week at ratios of 1:3 (young) and 1:2 (aged). All experiments were performed using all 6 human fibroblast cell lines at least for three times in triplicates.

### 4.3. Production of Young and Aged Cell Secretomes

Fibroblasts isolated from young and aged individuals were seeded into Nunc™ 4-well multi-dishes at density of 1 × 10^5^ cells/cm^2^ and cultured in fibroblast standard culture medium for 24 h. On day 2, cells were extensively washed with PBS, and incubated with 0.25 mL/well of FCM without FBS in 5% CO_2_ at 37 °C. After 48 h of culture, young (yCM) and aged secretomes/conditioned media (aCM) were collected, centrifuged twice at 1000× *g* for 10 min, filtered through 0.2 μm filters (Sarstedt, Milan, Italy), and stored at −80 °C until use.

### 4.4. EV Isolation/Depletion

EVs were purified from yCM and aCM using the qEVs 35 nm Size Exclusion Chromatography (SEC) columns (iZon), following the manufacturer’s instructions. Briefly, columns were left at room temperature for 30 min, washed with 2 volumes (13.5 mL each) of filtered PBS, equilibrated with 3 volumes (13.5 mL each) of filtered MEM, and injected with 1 mL of CM. EVs were collected and used for culture experiments (See Section 4.5), or subjected to quantification analysis, transmission electron microscopy (TEM), Western Blot, and miR-200b and miR200c expression studies. EV-depleted yCM (EV-yCM) were also collected and used for the experiments described in Section 4.5 and Section 4.7.

### 4.5. Aged Fibroblast Exposure to Cell Secretomes Using 2D Systems

Fibroblasts were seeded at a concentration of 7 × 10^4^ cells/cm^2^ in 4-well multi-dishes (Nunc) and allowed to attach overnight. The next day, the FCM was removed, cells were washed in PBS and exposed to yCM, and yCM-derived EVs were resuspended in fibroblast standard culture medium without FBS (MEM + yCM-derived EVs), EV-depleted yCM (EV-yCM), and aCM for 48 h. At the end of treatment, cells were returned to FCM and maintained in 5% CO_2_ for 10 days at 37 °C. Medium was replaced every two days. Cell morphology was monitored daily using an Eclipse TE200 inverted microscope (Nikon, Tokyo, Japan), which was connected to a Digital Sight camera (Nikon, Tokyo, Japan). Cultures were arrested at days 2, 5, 7, and 10, and analyzed as described below.

### 4.6. Generation of Young and Aged 3D ECM-Based Bio-Scaffolds

Ovaries from 6-month (young, *n* = 3) and 5-year-old (aged, *n* = 3) sows were subjected to the decellularization protocol previously developed in our laboratory [34,35,36,37]. Briefly, organs were frozen for at least 24 h at −80 °C, thawed, and treated with 0.5% sodium dodecyl sulfate (Bio-Rad, Milan, Italy) for 3 h, 1% Triton X-100 (Sigma-Aldrich, Milan, Italy) for 16 h, and 2% deoxycholate (Sigma-Aldrich, Milan, Italy) overnight. At the end of the procedures, the generated bio-scaffolds were extensively washed in double-distilled water (ddH_2_O) for 6 h, sterilized with 70% ethanol supplemented with 2% antibiotic/antimycotic solution (Sigma-Aldrich, Milan, Italy) for 30 min, and used as cell culture supports. Small fragments from each bio-scaffold were analyzed to evaluate the efficacy of the decellularization process and the composition of the model.

### 4.7. Aged Fibroblast Exposure to Cell Secretomes Using Young and Aged 3D ECM-Based Bio-Scaffolds

7 × 10^4^ aged fibroblasts/cm^2^ were seeded onto young and aged 3D ECM-based bio-scaffolds and allowed to attach overnight. The next day, the FCM was removed, and cells were washed in PBS and exposed to yCM or EV-yCM for 48 h. At the end of treatment, cells were returned to standard culture medium and maintained in 5% CO_2_ for 10 days at 37 °C. Medium was replaced every two days. EV-yCM was used as a control. Cultures were arrested at days 2, 5, 7, and 10, and embedded in paraffin for histological analyses or used for DNA quantification or gene expression studies.

### 4.8. Cell Proliferation Index

Cells cultured onto 2D systems were fixed in methanol for 15 min at −20 °C. Cells cultured onto 3D ECM-based bio-scaffolds were fixed in 10% neutral buffered formalin for 24 h, gradually dehydrated in alcohols, cleared with xylene, and embedded in paraffin. A total of 5-μm thick microtome sections were cut and rehydrated, and antigens were unmasked with 10 mM Sodium citrate solution (pH 6) containing 0.05% Tween-20 (Sigma-Aldrich, Milan, Italy) at 120 °C.

Non-specific cross-reacting antigens were blocked with 10% Goat Serum (Sigma-Aldrich, Milan, Italy) in PBS for 30 min. Anti-proliferating cell nuclear antigen (PCNA) primary antibody (1:200, Sigma-Aldrich, Milan, Italy) was incubated for 1 h, followed by secondary antibody exposure (1:250, Alexa Fluor™ 594) for 30 min. Nuclei were counterstained with 1 µg/mL 4′,6-diamidino-2-phenylindole (DAPI, Sigma-Aldrich, Milan, Italy). All steps were performed at room temperature, unless otherwise specified. At the end of the immunostaining, 2D cultured samples were observed under the Eclipse TE200 microscope (Nikon, Tokyo, Japan), while paraffin-embedded tissues were analyzed through a Leica DMR microscope; both microscopes were equipped with a Nikon digital camera. Pictures were acquired using NIS-Elements Software (Version 4.6; Nikon, Tokyo, Japan). The number of PCNA+ cells was counted in 10 randomly selected fields at 100× total magnification and expressed as a percentage of the total cell counted.

### 4.9. β-Galactosidase (β-GAL) and Reactive Oxygen Species (ROS) Activities

Human human galactosidase beta (β-GAL; MyBioSource, MBS721441, Milan, Italy) and reactive oxygen species (ROS; MyBioSource, MBS166870, Milan, Italy) ELISA kits were used to detect cell activities. After 20 min of sonication at 20 kHz, samples were centrifuged at 3000 rpm for 20 min, and supernatants were collected and analyzed at 450 nm using a Multiskan FC. Standard curves were generated via plotting the target concentrations versus absorbances, linear regression analysis was computed, and β-GAL and ROS fold changes were quantified and reported with the higher expression set to 1, while the other expression was relative to this value.

### 4.10. Histological Analysis and Cell Density Evaluation

Cells cultured onto 3D ECM-based bio-scaffolds were fixed in 10% neutral buffered formalin for 24 h, gradually dehydrated in alcohols, cleared with xylene, and embedded in paraffin. In total, 5-μm thick microtome sections were cut, rehydrated, and stained with hematoxylin (Histoline, Milan, Italy) and eosin (BioOptica, Milan, Italy) or with 1 µg/mL DAPI (Sigma-Aldrich, Milan, Italy).

Cell density was estimated via analyzing 15 DAPI-stained sections (5-µm-thick) obtained from each experimental group.

All samples were visualized through a DMR microscope (Leica, Milan, Italy) equipped with a digital camera (Nikon; Tokyo, Japan). Pictures were acquired using NIS-Elements Software (Version 4.6; Nikon, Tokyo, Japan) and constant exposure parameters. Five randomly selected fields for each section at 100X magnification were analyzed using the Manual Cell Counter tool (ImageJ software version 1.53j, https://imagej.nih.gov/ij/notes.html, accessed on 1 May 2023). Cell density was expressed as cell/mm^2^ of tissue.

### 4.11. DNA Quantification

Genomic DNA was extracted from repopulated bio-scaffold fragments using the PureLink^®^ Genomic DNA Kit, following the manufacturer’s instructions. DNA concentrations were quantified with NanoDrop 8000 and normalized against the previously annotated tissue weights.

### 4.12. Gene Expression Analysis

RNA was extracted using the TaqMan™ Gene Expression Cells to Ct kit. DNase I was added in lysis solutions at 1:100 concentration. The expression of target genes was analyzed using the CFX96 Real-Time PCR detection system (Bio-Rad Laboratories, Milan, Italy) and pre-designed primers and probe sets from TaqMan™ Gene Expression Assays (Table 1). GAPDH and ACTB were used as internal reference genes. Gene expression levels were quantified with CFX Manager software (Bio-Rad Laboratories, Milan, Italy), and reported with the highest expression set to 1, while the other expression was relative to this value.

### 4.13. Transmission Electron Microscope Analysis

Morphology of yCM- and aCM-derived EVs was analysed using TEM. Five microliters of EV suspensions were placed on a glow-discharged Formvar-Carbon Copper grid of 300 mesh (Sigma-Aldrich, Milan, Italy) for 2 min and stained with 2% uranyl acetate (Sigma-Aldrich, Milan, Italy) for a further 2 min. Samples were then washed with 0.1 µm filtered PBS for 1 min at room temperature. After drying, they were observed under a TEM Talos L120C at 120 KV. Images were acquired using a Ceta camera 4kx4k.

### 4.14. Western Blot

yCM- and aCM-derived EVs were lysed for 30 min at 4 °C in RIPA buffer (20 nM Tris-HCl, 150 nM NaCl, 1% deoxycholate, 0.1% SDS, 1% Triton X-100, pH 7.8), supplemented with protease and phosphatase inhibitors cocktail (Sigma-Aldrich, Milan, Italy). Protein concentrations were quantified using a BCA Protein Assay Kit. A total of 10 μg of proteins were loaded, electrophoresed on precast 4–20% polyacrylamide Mini-PROTEAN TGX gels (#4561093, Bio-Rad Laboratories, Milan, Italy) at 200V for 35 min, transferred onto nitrocellulose membranes (Hybond-C Extra, GE Healthcare Life Sciences, Milan, Italy), and probed with primary antibodies for CD9 (1:250, #555370, BD Biosciences, Milan, Italy), CD63 (1:1000, #ab271286, Abcam, Cambridge, UK), CD81 (1:1000, #ab79559, Abcam, Cambridge, UK) and beta Actin (1:5000, #ab6276, Abcam, Cambridge, UK). Protein bands were visualized using the WesternBreeze chemiluminescent kit, and densitometric analysis was performed using the ImageJ software (ImageJ software version 1.53j, https://imagej.nih.gov/ij/notes.html, accessed on 1 May 2023).

### 4.15. EV Quantification

EVs isolated from yCM and aCM were quantified using the ExoCET^®^ Exosome Quantitation Kit (System Biosciences, Milan, Italy), following the manufacturer’s instructions. Briefly, EVs were lysed, centrifuged at 1500× *g* for 5 min, and the supernatants were transferred in the microtiter plate for quantification at 405 nm using a Multiskan FC. The standard curve was obtained through plotting the target concentrations versus absorbances. Linear regression analysis was computed, and EV quantifications were calculated and reported with the highest expression set to 1, while the other expression was relative to this value.

### 4.16. miR-200b and miR-200c Expression Analysis

Small RNAs were extracted from yCM- and aCM-derived EVs using the Total Exosome RNA and Protein Isolation Kit, following the provider’s instructions. miR-200b and miR-200c were selectively reverse-transcribed using TaqMan™ MicroRNA Reverse Transcription kit and TaqMan™ pre-designed probes (Mir200b #002251; Mir200c #002300). RT-qPCR was performed with Universal Master Mix II and TaqMan™ microRNA Assay (Mir200b #002251; Mir200c #002300), using the CFX96 Real-Time PCR detection system (Bio-Rad Laboratories, Milan, Italy). miR-200b and miR-200c contents are here reported; the highest expression was set to 1, while the other was relative to this value.

### 4.17. Statistical Analysis

Statistical analysis was performed using two-way ANOVA (SPSS 19.1; IBM). Data were reported as the mean ± standard deviation (SD). Differences of *p*  ≤  0.05 were considered significant and indicated with different lowercase letters.

## Figures and Tables

**Figure 1 ijms-24-08285-f001:**
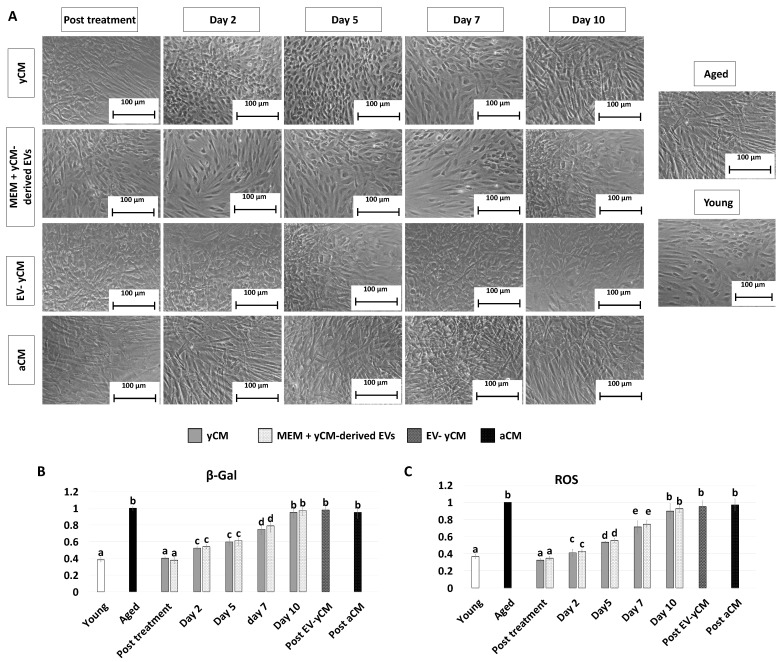
Morphology, β-GAL, and ROS activity quantifications in treated aged fibroblasts cultured onto 2D systems. (**A**) Representative images of untreated fibroblasts isolated from aged (black, Aged) and young donors (white, Young); and aged fibroblasts at the end of 48-h exposure (post-treatment) to yCM, MEM + yCM-derived EV, and EV-depleted yCM (EV-yCM) and aCM at days 2, 5, 7, and 10 of culture. Scale bars: 100 µm. (**B**) β-GAL activity and (**C**) ROS levels in untreated fibroblasts isolated from young (Young) and aged donors (Aged); aged fibroblasts at the end of yCM and MEM + yCM-derived EV treatments (post-treatment) at days 2, 5, 7, and 10 of culture; and aged fibroblasts exposed to EV-depleted yCM (EV-yCM) and aCM for 48 h. Data are expressed as the mean. Error bars represent the standard error of the mean (SEM). Different lowercase letters indicate *p* < 0.05.

**Figure 2 ijms-24-08285-f002:**
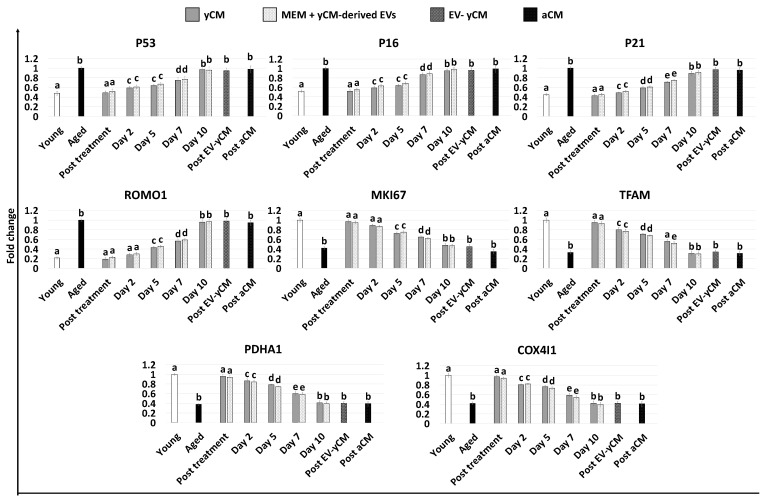
Gene expression levels of senescence-related markers (P53, P16, P21), reactive oxygen species modulator (ROMO1), cell proliferation (MKI67), and mitochondrial activity (TFAM, PDHA1, COX4I1) genes in untreated fibroblasts isolated from young (white, Young) and aged (black, Aged) individuals, at the end of yCM and MEM + yCM-derived EV exposures (Post treatment), at days 2, 5, 7, and 10 of culture, and in aged cells exposed to EV-depleted yCM (EV-yCM) and aCM for 48 h. Data are expressed as the mean. Error bars represent the standard error of the mean (SEM). Different lowercase letters indicate *p* < 0.05.

**Figure 3 ijms-24-08285-f003:**
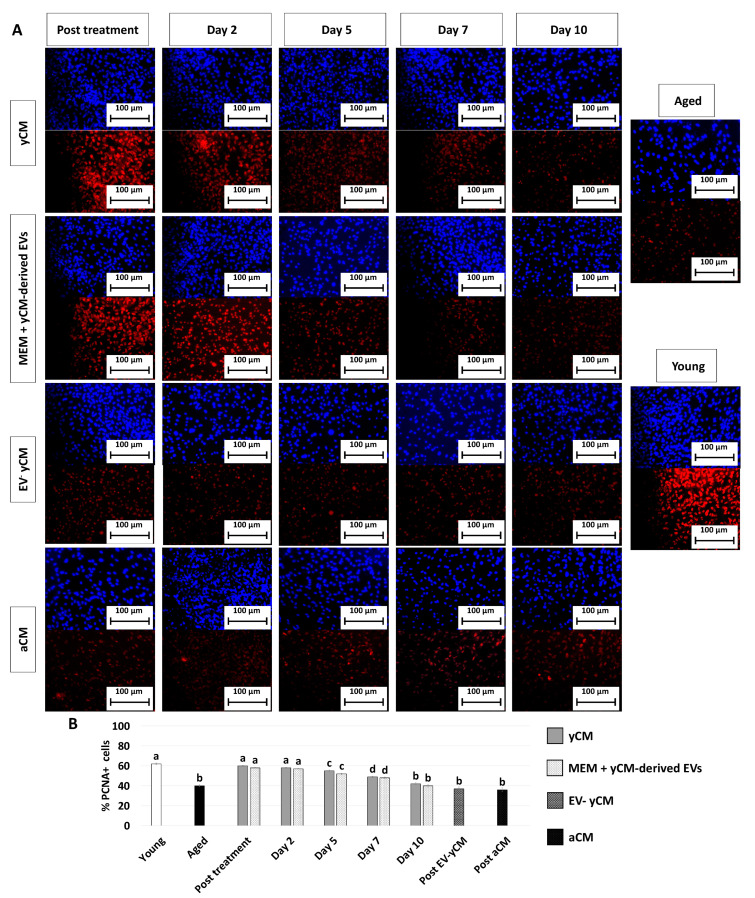
PCNA immunocytochemical staining and quantitative evaluation. (**A**) Representative images of DAPI (blue, upper panels) and PCNA (red, lower panels) immunostaining in untreated fibroblasts isolated from aged (black, Aged) and young donors (white, Young), and of aged fibroblasts at the end of 48-h exposure period (post-treatment) to yCM, MEM + yCM-derived EV, EV-depleted yCM (EV-yCM), and aCM at days 2, 5, 7 and 10 of culture. Scale bars: 100 µm. (**B**) PCNA quantitative evaluation in untreated fibroblasts isolated from young (Young) and aged individuals (Aged), at the end of yCM and MEM + yCM-derived EV exposures (post-treatment), at days 2, 5, 7, and 10 of culture, and in aged cells exposed to EV-depleted yCM (EV-yCM) and aCM for 48 h. Data are expressed as the mean. Error bars represent the standard error of the mean (SEM). Different lowercase letters indicate *p* < 0.05.

**Figure 4 ijms-24-08285-f004:**
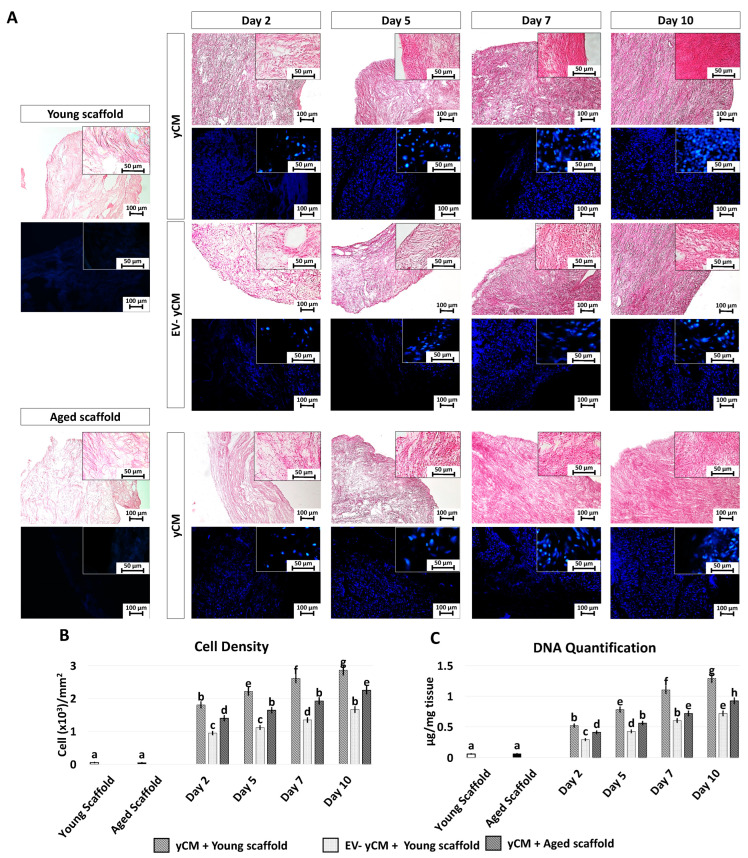
H and E, DAPI staining, cell density, and DNA quantification in treated aged fibroblasts cultured onto young and aged 3D ECM-based bio-scaffolds. (**A**) H and E (upper panels) and DAPI (blue, lower panels) staining of young (white, Young scaffold) and aged ECM-based bio-scaffolds (black, Aged scaffold), aged fibroblasts exposed to yCM and EV-depleted yCM (EV-yCM) cultured onto young scaffold, and aged fibroblasts exposed to yCM and engrafted onto aged scaffolds at days 2, 5, 7, and 10 of culture. Scale bars: 100 µm. Insert scale bars: 50 µm. (**B**) Cell density analyses in young and aged scaffolds before cell seeding, aged fibroblasts exposed to yCM and EV-yCM cultured onto young scaffolds (yCM + Young scaffold, EV-yCM + Young scaffold), and aged fibroblasts exposed to yCM and engrafted onto aged scaffolds (yCM + Aged scaffold) at days 2, 5, 7, and 10 of culture. Data are expressed as the mean. Error bars represent the standard error of the mean (SEM). Different lowercase letters indicate *p* < 0.05. (**C**) DNA quantification in young and aged scaffolds before cell seeding, aged fibroblasts exposed to yCM and EV-yCM cultured onto young scaffolds (yCM + Young scaffold, EV-yCM + Young scaffold), and aged fibroblasts exposed to yCM and engrafted onto aged scaffolds (yCM + Aged scaffold) at days 2, 5, 7, and 10 of culture. Data are expressed as the mean. Error bars represent the standard error of the mean (SEM). Different lowercase letters indicate *p* < 0.05.

**Figure 5 ijms-24-08285-f005:**
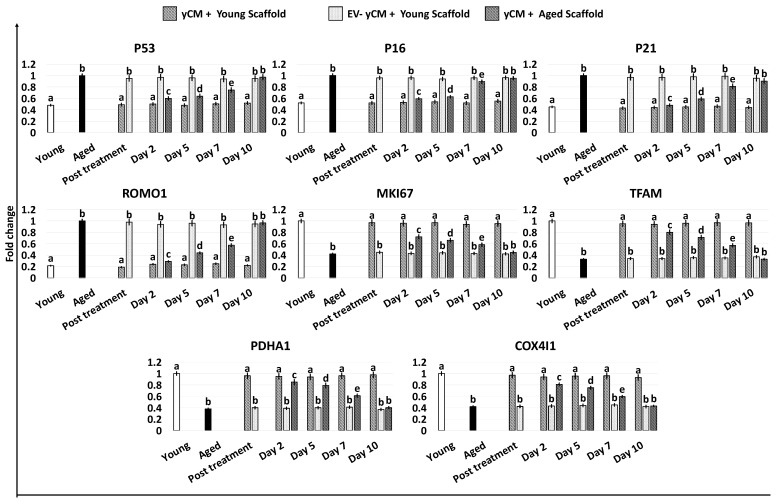
Gene expression levels of senescence-related markers (P53, P16, P21), reactive oxygen species modulator (ROMO1), cell proliferation (MKI67), and mitochondrial activity (TFAM, PDHA1, COX4I1) genes in untreated fibroblasts isolated from young (white, Young) and aged individuals (black, Aged) at the end of yCM and EV-depleted yCM (EV-yCM) treatments (post-treatment), and in yCM + Young scaffold, EV-yCM + Young scaffold, and yCM + Aged scaffold at days 2, 5, 7, and 10 of culture. Data are expressed as the mean. Error bars represent the standard error of the mean (SEM). Different lowercase letters indicate *p* < 0.05.

**Figure 6 ijms-24-08285-f006:**
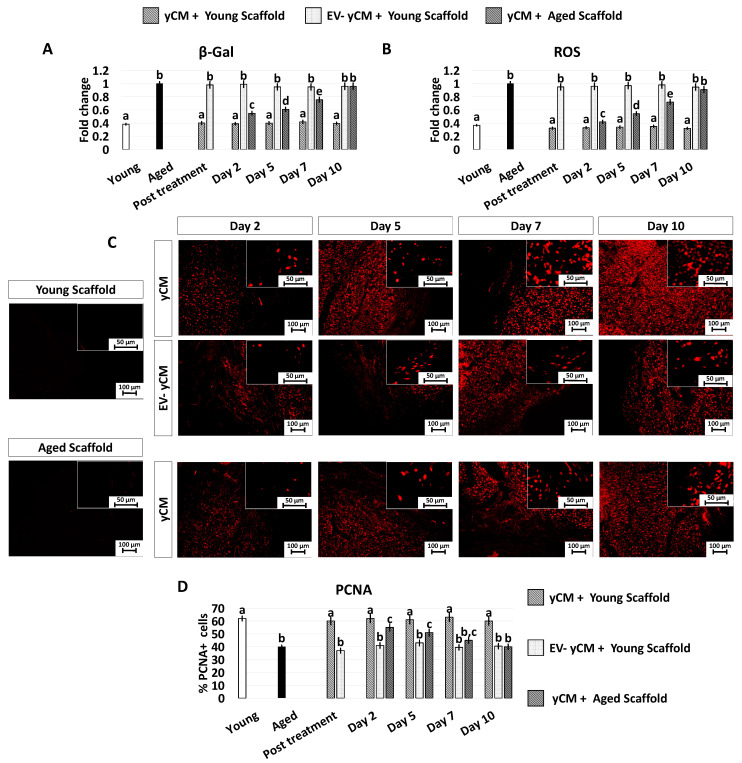
β-GAL and ROS activity quantifications, PCNA immunocytochemical staining, and quantitative evaluation in treated aged fibroblasts cultured onto young and aged 3D ECM-based bio-scaffolds. (**A**) β-GAL activity and (**B**) ROS levels in untreated fibroblasts isolated from young (white, Young) and aged individuals (black, Aged), aged fibroblasts exposed to exposed to yCM and EV-depleted yCM (EV-yCM) cultured onto young scaffolds (yCM + Young scaffold, EV-yCM + Young scaffold), and aged fibroblasts exposed to yCM and engrafted onto aged scaffolds (yCM + Aged scaffold) at days 2, 5, 7, and 10 of culture. Data are expressed as the mean. Error bars represent the standard error of the mean (SEM). Different lowercase letters indicate *p* < 0.05. (**C**) Representative images of PCNA immunocytochemical staining (red) of young (Young scaffold) and aged ECM-based bio-scaffolds (Aged scaffold) before seeding, aged fibroblasts exposed to yCM and EV-depleted yCM (EV-yCM) cultured onto young scaffolds (yCM + Young scaffold, EV-yCM + Young scaffold), and aged fibroblasts exposed to yCM and engrafted onto aged scaffolds (yCM + Aged scaffold) at days 2, 5, 7, and 10 of culture. Scale bars: 100 µm. Insert scale bars: 50 µm. (**D**) PCNA quantitative evaluation in untreated fibroblasts isolated from young (Young) and aged individuals (Aged), aged fibroblasts exposed to exposed to yCM and EV-depleted yCM (EV-yCM) cultured onto young scaffolds (yCM + Young scaffold, EV-yCM + Young scaffold), and aged fibroblasts exposed to yCM and engrafted onto aged scaffolds (yCM + Aged scaffold) at days 2, 5, 7, and 10 of culture. Data are expressed as the mean. Error bars represent the standard error of the mean (SEM). Different lowercase letters indicate *p* < 0.05.

**Figure 7 ijms-24-08285-f007:**
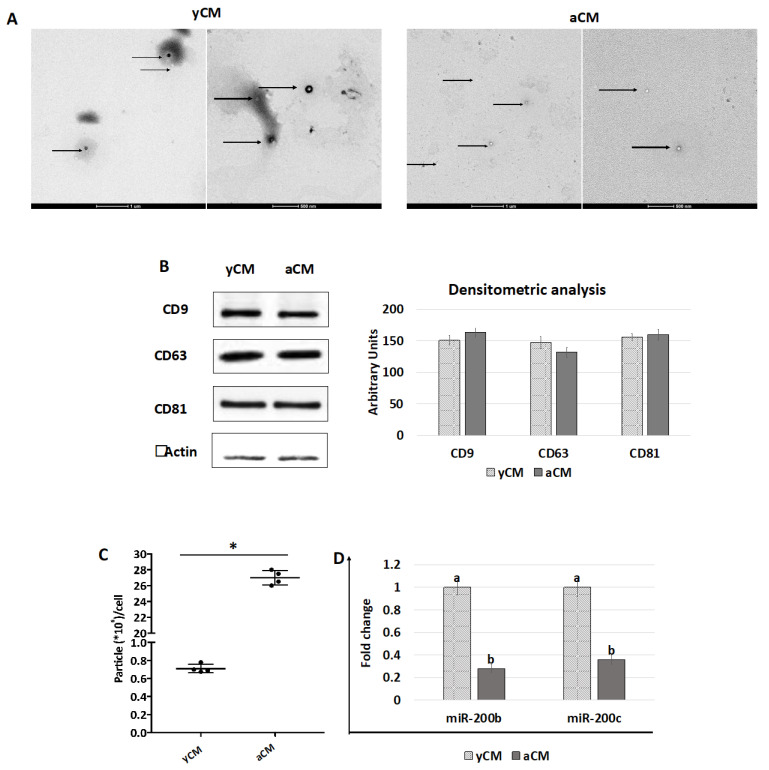
TEM, western blot, quantification, and miR200b and miR200c expression studies in yCM and aCM. (**A**) Representative images of TEM analysis of yCM- and aCM-derived EVs. Black arrows indicate spherical structures displaying the distinctive EV cup morphology, surrounded by a lipid bilayer with variable electron density cargo content. Scale bars 1 µm and 500 nm. (**B**) Western blot and densitometric analyses (arbitrary units) of EV proteins for the tetraspanin markers CD9, CD63, CD81, and beta Actin in yCM and aCM. (**C**) Quantification of EV particles detected in yCM and CM. Data are expressed as the mean. Error bars represent the standard error of the mean (SEM). * indicate *p* < 0.05. (**D**) Expression levels of miR-200b and miR-200c in yCM- and aCM-derived EVs. Data are expressed as the mean. Error bars represent the standard error of the mean (SEM). Different lowercase letters indicate *p* < 0.05.

**Table 1 ijms-24-08285-t001:** List of primers used for quantitative PCR analysis.

Gene	Description	Cat. N.
ACTB	Actin, beta	Hs01060665_g1
GAPDH	Glyceraldehyde-3-phosphate dehydrogenase	Hs02786624_g1
MKI67	Marker of proliferation Ki-67	Hs04260396_g1
TFAM	Transcription Factor A, Mitochondrial	Hs00273372_s1
ROMO1	Reactive oxygen species modulator 1	Hs00603977_m1
PDHA1	Pyruvate dehydrogenase E1 subunit alpha 1	Hs01049345_g1
COX4I1	Cytochrome C oxidase subunit 4I1	Hs00971639_m1
CDKN1 (P21)	Cyclin-dependent kinase inhibitor 1A	Hs00355782_m1
CDKN2A (P16)	Cyclin-dependent kinase inhibitor 2A	Hs00923894_m1
TP53 (P53)	Tumor protein p53	Hs01034249_m1

## Data Availability

The data presented in this study are available on request from the corresponding author.

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
