# Peer review of "3D ECM-Based Scaffolds Boost Young Cell Secretome-Derived EV Rejuvenating Effects in Senescent Cells"

_ijms, 2023, doi:10.3390/ijms24098285_

Round 1

Reviewer 1 Report

Article present strong data, with detailed result with a lot of material proving the quality of data. I strong recommend to publish this article in this present form.

Author Response

We are grateful to the Reviewer for her/his appreciation.

Reviewer 2 Report

The authors investigated the effects and the content of young cell secretome-derived EVs. They also studied the use of decellularized bio-scaffolds as extracellular matrix (ECM)-based supports which maintains the rejuvenated phenotype in senescent cells treated with yCM-derived EVs.  

Today, this topic on secretome and EV is original as well as the studies on aging are too. 

Therefore, in my opinion, the paper is well organized, the authors used opportune methodologies for the characterization of the cells treated with EVs. The conclusions are consistent with their results and scientific evidence. The references are appropriate, as well as the figures. 

The paper can be accepted in the present form. 

Minor editing of English language required

Author Response

We thank the Reviewer for appreciating the quality of the experiments described and for her/his positive evaluation of the manuscript. English language has been carefully edited throughout the text.

Reviewer 3 Report

This manuscript characterized the role of secretome from young and aged fibroblasts as well as extracellular matrix (ECM) scaffolds in rejuvenating senescent cells. The authors found that the extracellular vesicles (EVs) from young donors can revert the senescence phenotype of fibroblasts compared to aged donors. The effects disappear if EVs were removed from the media but can be stabilized in ECM scaffolds. This is a well designed and executed study, with great integration of various cell studies, 3D cultures, and compelling observations. 

  1. In the introduction, the references for miR-200b and miR-200c should be added to better motivate the study. Similarly, the discussion only briefly mentions the miR-200 family without references. How miR-200 family and increased EV secretion affects the young condition medium and its role in reverting senescence were not examined in this manuscript. The authors should expand the discussion in more detail in this area. 

  2. The mechanical properties of young and aged ECM were not characterized and the authors should not claim the effects were due to their mechanical microenvironment as there are many factors that can influence the outcome.  

  3. In the methods section, the authors described the use of human primary skin fibroblast cell lines (GM02674, GM00495, GM08402, GM01706, GM00731, and AG09602) but how were they used in this manuscript? If the authors claim to use all 6, then the representative images and associated quantification from each cell line should be shown and specified.

  4. In figure 6C, the scaffold before seeding still has some baseline fluorescence of PCNA and many images on the right side seem to have nonspecific staining too. It is not uncommon for tissue staining but it will make the arguments stronger if the authors can use the scaffold images to threshold positive signals in the treatment groups with cell seeding. 

  5. In figure 7B, the western blots should show the control and quantification. 

Author Response

This manuscript characterized the role of secretome from young and aged fibroblasts as well as extracellular matrix (ECM) scaffolds in rejuvenating senescent cells. The authors found that the extracellular vesicles (EVs) from young donors can revert the senescence phenotype of fibroblasts compared to aged donors. The effects disappear if EVs were removed from the media but can be stabilized in ECM scaffolds. This is a well designed and executed study, with great integration of various cell studies, 3D cultures, and compelling observations. 

We thank the Reviewer for her/his appreciation and for her/his suggestions that improve the quality of the manuscript.

1. In the introduction, the references for miR-200b and miR-200c should be added to better motivate the study. Similarly, the discussion only briefly mentions the miR-200 family without references. How miR-200 family and increased EV secretion affects the young condition medium and its role in reverting senescence were not examined in this manuscript. The authors should expand the discussion in more detail in this area. 

As suggested by the Reviewer, the references for miR-200b and miR-200c were added in the introduction (please see line 75, Ref 3 and 21).

We fully agree with the Reviewer’s observation that further studies are needed to in-depth elucidate the role of miR-200 family and increased EV secretion in young condition medium and how all these interactions result in reverting senescence. However, we feel that this is beyond the scope of the present manuscript which rather focuses on the possible synergies existing between 3D ECM-based scaffolds and young cell secretome-derived EVs. Nevertheless, a paragraph and related references were added to the discussion in order to further examine the role of miR-200 family in reverting senescence (please see lines 346-357).

 “In agreement with this hypothesis, miR-200 family ability to restore normal functions in different senescent cell types has been previously demonstrated [3, 21]. A clear example can be found in idiopathic pulmonary fibrosis which causes a down regulation of miR-200 members in alveolar epithelial cells, leading to the acquisition of a senescent phenotype [32,33]. It is, however, interesting to note that miR-200b and miR-200c transfection restores cell transdifferentiation ability and induces a significant reduction of aging hallmarks in senescence alveolar cells [21]. Similarly, the use of miR-200b and miR-200c has been shown to directly regulate the molecular mechanisms erasing cell senescence in fibroblasts isolated from old individuals [3]. Although further studies are needed to better elucidate these aspects, based on all these observations, we may speculate that young cells release EVs containing high amount of miR-200s, required to support normal, physio-logical cell functions, and that this ability decreases in aged as well as in pathological cells.”

2. The mechanical properties of young and aged ECM were not characterized and the authors should not claim the effects were due to their mechanical microenvironment as there are many factors that can influence the outcome.  

As suggested by the Reviewer, the term “mechanical” was deleted and more general expression were used throughout the text (please see lines 26, 28, 361).

3. In the methods section, the authors described the use of human primary skin fibroblast cell lines (GM02674, GM00495, GM08402, GM01706, GM00731, and AG09602) but how were they used in this manuscript? If the authors claim to use all 6, then the representative images and associated quantification from each cell line should be shown and specified.

We thank the Reviewer for this suggestion. However, we would like to draw her/his attention to the fact that, during the preliminary studies aimed to characterize the 6 cell lines and to setup of the protocols, we did not find statistically significant differences among the 3 cell lines isolated from aged individuals (GM01706, GM00731, AG09602), nor among the 3 cell lines obtained from young donors (GM02674, GM00495, GM08402). Based on this, GM01706, GM00731, AG09602 lines were used as biological replicates for the “aged” group and GM02674, GM00495, GM08402 for “young” group.

4. In figure 6C, the scaffold before seeding still has some baseline fluorescence of PCNA and many images on the right side seem to have nonspecific staining too. It is not uncommon for tissue staining but it will make the arguments stronger if the authors can use the scaffold images to threshold positive signals in the treatment groups with cell seeding. 

As suggested by the Reviewer, we used the scaffold images to threshold positive signals in the treatment groups with cell seeding.  The new version of Figure 6C is now included in the revised manuscript (please see page 8).

5. In figure 7B, the western blots should show the control and quantification. 

As suggested by the Reviewer, western blot control and quantification were added in figure 7B and included in the new version of the manuscript (please see page 9). Figure legend 7 and Material and method section were modified accordingly (please see lines 260-261 and 519-521).
